# Relationship between Racial Diversity in Medical Staff and Hospital Operational Efficiency: An Empirical Study of 3870 U.S. Hospitals

**DOI:** 10.3390/bs13070564

**Published:** 2023-07-06

**Authors:** C. Christopher Lee, Young Sik Cho, Diosmedy Breen, Jessica Monroy, Donghwi Seo, Yong-Taek Min

**Affiliations:** 1School of Business, Central Connecticut State University, New Britain, CT 06050, USA; christopher.lee@ccsu.edu (C.C.L.); burgos@my.ccsu.edu (D.B.); jm8856@ccsu.edu (J.M.); 2College of Business, Jackson State University, Jackson, MS 39217, USA; 3Lubin School of Business, Pace University, New York, NY 10038, USA; jseo@pace.edu; 4Department of Health Sciences, Florida Gulf Coast University, Fort Myers, FL 33965, USA; ytmin@fgcu.edu

**Keywords:** workforce racial diversity, hospital operational efficiency, ESG management, DEI, foreign nurses, healthcare operations management, stakeholder theory, contingency theory, COVID-19 pandemic, empirical study

## Abstract

Demand for foreign nurses and medical staff is rapidly increasing due to the severe labor shortage in U.S. hospitals triggered by the COVID-19 pandemic. However, empirical studies on the effect of the racial diversity of medical staff on hospital operations are still lacking. This research gap is thus investigated based on the foreign medical staff working in 3870 U.S. hospitals. Results show that workforce racial diversity has a significantly positive relationship with hospital operational efficiency regarding occupancy rate, manpower productivity, capacity productivity, and case mix index. Notably, this study empirically supports that increasing the ratio of foreign nurses positively affects the overall operational efficiency of hospitals. In addition, the study results also indicate that the hospital location, size, ownership, and teaching status act as significant control variables for the relationship between racial diversity and hospital efficiency. These results imply that hospitals with these specific operating conditions need to pay more attention to racial diversity in the workplace, as they are structurally more sensitive to the relationship between racial diversity and operational efficiency. In short, the findings of this study suggest that hospital efficiency can be operationally improved by implementing workforce ethnic diversity. For this reason, hospital stakeholders and healthcare policymakers are expected to benefit from this study’s findings. Above all, the results of this study imply that if an organization adapts to extreme external environmental changes (e.g., the COVID-19 pandemic) through appropriate organizational restructuring (i.e., expanding the workforce racial diversity by hiring foreign medical staff), the organization can gain a competitive advantage, a claim that is supported by contingency theory. Further, investors are increasingly interested in ESG, especially companies that embody ethical and socially conscious workplaces, including a diverse and inclusive workforce. Thereby, seeking racial diversity in the workforce is now seen as a fundamental benchmark for organizational behavior that predicts successful ESG business practices, a claim that is supported by stakeholder theory. Therefore, in conclusion, the findings of this study suggest that workforce racial diversity is no longer an optional consideration but should be considered as one of the essential determinants of competitive advantage in organizations, particularly in the healthcare sector.

## 1. Introduction

### 1.1. COVID-19 Pandemic and Workforce Racial Diversity

McKinsey’s research [1] reported that diversity and inclusion were at risk globally during the COVID-19 pandemic, but that a diverse workforce is more critical than ever for business recovery and resilience. When the McKinsey Global Institute (n = 1122 executives globally) surveyed the extent to which CEOs made Diversity, Equity, and Inclusion (DEI) a priority before and during COVID-19, 9 out of 10 said DEI was moderate to very high despite the pressures of the COVID-19 crisis; specifically, 29% said it was a top priority and 40% said it was a very important priority [2]. In fact, companies have been aware of the positive aspects of diversity for years, as shown by Hansen [3] who estimated that companies spend nearly $10 billion annually on DEI. Despite enormous spending on DEI, minorities constituted 33% of the U.S. population but only 14% of all top-level managers and officials in private U.S. hospitals [4]. This represents a major shortcoming in U.S. hospitals, as diversity was positively associated with both increased patient care and financial performance [5]. Previous research has also shown that companies that invest in diversity and inclusion are better positioned to gain a competitive advantage by building more adaptable and effective teams [1].

In addition, efficiency is considered the key to success in any business. This also applies to organizations outside of traditional business structures, such as hospitals. Efficiency is becoming more critical in many healthcare systems worldwide. The challenge is meeting growing demand while facing resource constraints regarding staff and equipment [6]. Healthcare efficiency is imperative for refining equity, effectiveness, and access to healthcare for society [7], which is something echoed in Tiemann and Schreyögg’s [8] sentiment about how rising health expenditure and tight public budgets over the past four decades have led industrialized countries to seek ways to improve the performance of healthcare organizations. While research on workforce diversity in Fortune 500 companies continues [9], in the healthcare industry, research on the relationship between organizational behavior toward diversity and hospital performance is still lacking [10]. Especially, to the best of our knowledge, no empirical studies have directly investigated the effect of workforce racial diversity on hospital efficiency in the context of healthcare operations management. There are several reasons why this study has not been conducted yet. The biggest barrier to this kind of research appears to be the difficulty in collecting reliable large datasets and accurately measuring operational efficiency, specifically, in hospital operations. In addition, a large number of foreign nurses and medical staff are being hired due to the severe hospital staff shortage triggered by this turbulent time of the COVID-19 pandemic [11]. Accordingly, it is necessary to understand how the workforce racial diversity caused by the increase in foreign medical personnel affects hospital operational efficiency. Therefore, this study explores the relationship between the ethnic diversity of medical staff and the efficiency of hospital operations. In short, this study aims to answer the following research question: how does the racial diversity of medical staff affect hospital operational efficiency?

### 1.2. Research Outline

Sample data from the American Hospital Association (AHA) survey were employed to examine the research question empirically. This sample data covered most U.S. hospitals where information was available to measure operational efficiency metrics. In particular, the staff efficiency, occupancy rate, manpower productivity, capacity productivity, and case mix index were adopted to evaluate and compare the hospital operational efficiency of each U.S. hospital. Foreign nurse employment (FORNRSA) was used as a proxy variable to measure the level of workforce racial diversity. According to the AHA data, a total of 694 U.S. hospitals were classified into the high-diversity group and 3176 U.S. hospitals into the low-diversity group.

In addition, previous research has argued that there are significant differences between hospital types and strategic initiatives to promote diverse values and practices [7]. Hence, this study also investigated which factors were influential as control variables in the relationship between workforce ethnic diversity and hospital operational efficiency. Based on the literature review, the following variables were adopted as control variables to be investigated: hospital ownership (government, not-for-profit, and for-profit), hospital location (rural, microcity, and metropolitan), hospital size (small, medium, and large), and teaching status (teaching vs. non-teaching). 

Finally, based on the findings of the data analysis, theoretical and managerial implications are discussed in the context of healthcare operations management. We also report the limitations of the study and future research directions.

## 2. Theoretical Backgrounds and Hypotheses

### 2.1. Literature Review 

Homroy and Soo [12] studied the effect of diversity in groups regarding both group performance and individual performance post-working in diverse groups. The study distinguished diversity in three ways: nationality, gender, and ability. They ran their experiment on students in a real-life academic setting where they received actual grades for their work. The findings were that no group diversity of any of the three types of diverse groups was statistically significant. The experiment then further examined individual performances after the group performances were complete. This study found that working in nationally diverse teams did have small but statistically significant effects on students’ performances. The research suggests that individuals who can work in nationally diverse teams and learn from being part of a team with diverse team members may lead to better individual performance. Blouch and Malik [13] studied the relationship between employee perceptions of diversity (EPD) and perceived organizational performance (POP) across the private and public healthcare industry through the intervention of the employee perception of justice in Pakistan. Medical information was collected from the private and public sectors to analyze the results. Surveys were administered to 250 participants, and hypotheses were tested using two-way ANOVA and CAUSALMED procedures. The findings showed that employees had an increase in their perception toward organizational performance by about one-third if they perceived diversity-related initiatives to maintain and value a diverse workforce within the organization. The study provides strong insights into employees’ behavior related to diversity and its impact on organizational performance in the presence of perceived organizational justice (POJ). Singal [14] researched whether investing in diversity management and diversity performance contributed to higher business performance. The sample data covered a period of 20 years and 35,416 firm-years’ worth of data as well as financial information used to compute Tobin’s Q, leverage, and firm size. They used regression equations to determine if there was a relationship between diversity performances in previous periods and the current period of financial performance. The study concluded that improvements made in the overall diversity performance positively impacted future financial performance [14]. Ye et al. [15] studied the relationship between leadership gender diversity on capital market efficiency, measured by stock liquidity in the Chinese A-share market. Although this study was on something outside of the healthcare sector, it focused on the positive correlations between gender diversity and efficiency and the difference in this correlation between gender-diverse leadership and non-gender-diverse leadership. They found that gender diversity had a positive correlation with efficiency, but this correlation was much more potent when gender diversity also existed in the board room. This finding demonstrated that top-level diversity is a prerequisite to achieving the benefits of workplace diversity. In addition to this financial benefit of workplace diversity in hospital operations, previous healthcare studies have also shown a clinical benefit that patients generally achieve better treatment outcomes when care is delivered by a diverse team [5].

### 2.2. Workforce Racial Diversity and Hospital Operational Efficiency

As discussed so far, previous studies have reported that workforce diversity is an essential factor influencing the effectiveness of different types of organizations. Accordingly, it is also reasonably expected that the diversity of the hospital workforce will have a positive impact on the efficiency of hospital operations. Prior research has indicated that organizations investing in diversity and inclusion are more likely to build highly adaptable and effective teams [1]. Further, investors are increasingly interested in ESG, which refers to corporate behavior on the three pillars of environment, society, and governance [16]. Socially conscious investors, especially, look for companies that promote ethical and socially conscious workplaces, including a diverse and inclusive workforce, from subcontractors to top management [17,18]. As a result, pursuing workforce diversity is seen as an essential corporate strategy for creating a competitive advantage today [1]. Hence, considering the severe shortage of hospital staff triggered by the turbulent COVID-19 pandemic, it is inferred that increasing the racial diversity of medical staff by recruiting competent foreign medical staff will improve hospital operational efficiency and help secure support from investors and stakeholders. 

This claim can be also supported by the contingency theory [19] as well as the stakeholder theory [20]. First, let us assume that a hospital solves the serious shortage of medical staff due to environmental conditions such as the COVID-19 pandemic by expanding the recruitment of overseas medical staff. From the contingency theory perspective, this organizational behavior can be interpreted as trying to secure survival and a sustainable competitive advantage by appropriately adapting their organizational structure to structurally changing environments [21].

Stakeholder theory asserts that managers must “manage and integrate the relationships and interests of shareholders, employees, customers, suppliers, communities, and other groups in a way that ensures the long-term success of the firm” [22] (p.10). The stakeholder approach emphasizes active management of the business environment, relationships, and promotion of shared interests [22] (p.10). From the stakeholder theory perspective, if hospitals gain shareholder support and promote common interests with staff and communities by increasing racial diversity in the workplace, this can be interpreted as successfully fulfilling corporate social responsibility, one of the core values of ESG management [23,24].

Therefore, it suggests that the racial diversity of the workforce has a positive (+) relationship with the effectiveness of hospital operations. Based on the argument so far, this study established the following hypotheses to empirically examine the relationship between the racial diversity of medical staff and hospital operational efficiency. The overall research framework is presented in Figure 1 and a detailed discussion of the other hypotheses follows in this section.

**Hypothesis** **1.***Workforce racial diversity is positively associated with hospital operational efficiency*.

### 2.3. Workforce Racial Diversity, Hospital Efficiency, and Hospital Size

White and Ozcan [25] revealed differences between church-owned and secular hospitals when input/output, including bed size variables, was used to measure diversity and hospital efficiency. Coyne et al. [26] studied whether hospital size affected the efficiency and cost results for hospitals in Washington. The sample size for this analysis was 96, and it categorized hospitals into three bed sizes: small (1–40 beds), medium (41–150 beds), and large (151 or more beds). The data were tested for differences using the ANOVA test and further examined using Scheffe’s method. The results indicated that size matters in efficiency and is highly significant. Thus, based on the literature, it is expected that the hospital size can control the effect of workforce racial diversity on hospital operational efficiency. Hence, the following hypothesis is suggested for empirical testing.

**Hypothesis** **2.***Hospital size will control the impacts of workforce racial diversity on hospital operational efficiency*.

### 2.4. Workforce Racial Diversity, Hospital Efficiency, and Hospital Location

Smith [27] studied the racial segregation of hospital care among Medicare beneficiaries. Data were obtained from the 1993 Medicare provider analysis and review file and the 1990 U.S. census summary. Linear regression models were used to test the impact of metropolitan measures on hospital segregation. Hospital segregation from 126 hospitals in metropolitan areas was positively related to population demographics (size and density) and negatively correlated to income and productivity. The finding concluded a significant change in the South (since the pre-civil rights era). This study suggested that metropolitan hospitals had a disproportionate share and were more diverse, however, they were more vulnerable to proposed changes in Medicare. Ghosh et al. [28] studied the relationship between the diversity of hospital patients and the length of stay in New York. Data were employed from the New York State Inpatient Database. The study was conducted on the length of stay of White, Black, and Hispanic patients and included other minorities. For Hispanic patients, the length of stay (LOS) was more significant in a hospital in New York, which had the most diverse population. However, after an adjustment was made, the study concluded that there was no significant difference. The study did not shed light on whether efficiencies in inpatient care in New York hospitals were measured by LOS and the diversity of hospitals based on minorities. Garcia-Lacalle and Martin [29] studied the performance of rural and urban hospitals in Spain based on efficiency and perceived quality. The sample size included 13 rural and 14 urban hospitals for a total of 27 hospitals. The methodology used was data envelopment analysis (DEA) to evaluate the efficiency and statistical analysis that included the Mann–Whitney test to compare the performance between the two. The results indicated that the efficiency between both urban and rural hospitals performed similarly. Sun et al. [30] researched whether a government hospital’s geographic location influenced operational and financial performance. The sample included 68 hospitals in eastern, central, and western China. The results indicated that a hospital’s location led to significant financial performance differences. Hospitals in eastern China had higher revenue and expenses because of the economic developments and growth of insurance programs. These previous studies indicated that hospital location (e.g., rural, microcity, and metropolitan) can control the effect of workforce racial diversity on hospital operational efficiency. Accordingly, we propose the following hypothesis for empirical verification.

**Hypothesis** **3.***Hospital location will control the impacts of workforce racial diversity on hospital operational efficiency*.

### 2.5. Workforce Racial Diversity, Hospital Efficiency, and Hospital Ownership

Jehu-Appiah et al. [31] analyzed the technical efficiency of a district hospital in Ghana. Hospital ownership included government, mission, private, and quasi-government district hospitals. The study used DEA analysis to estimate the efficiency of 128 district hospitals. The results revealed that quasi-government-owned hospitals had the highest mean efficiency score (83%). The study concluded that quasi-hospital ownership was positively correlated with hospital efficiency. Stock and McDermott [32] studied how the contextual factors of ownership type affected the cost performance of New York hospitals.

The sample size of this study included 209 hospitals. To test the hypothesis, a hierarchical regression analysis was used. The results indicated that cost performance did not have a significant relationship with ownership status. Tiemann and Schreyögg [33] evaluated the efficiency of public, private for-profit, and private nonprofit in German hospitals. The data were obtained between 2002 and 2006. The sample size consisted of 1046 hospitals. To determine efficiency, this study utilized the DEA model to compare whether organizations produced the most output mix based on their input mix or how much was produced based on their input mix. Then, a linear regression analysis was used to assess whether the ownership type affected the efficiency level. The results indicated that hospitals with public ownership had higher efficiency scores than the other types of hospitals used in this study. Based on these studies, it is rationally assumed that a hospital’s ownership status also can control the effect of workforce racial diversity on hospital operational efficiency. As such, the following hypothesis is proposed for empirical testing.

**Hypothesis** **4.***Hospital ownership will control the impacts of workforce racial diversity on hospital operational efficiency*.

### 2.6. Workforce Racial Diversity, Hospital Efficiency, and Teaching Status 

Messina et al. [34] sought to determine if there was a difference in the relationship between patient satisfaction and admissions across teaching and non-teaching hospitals. Data were collected from a survey questionnaire, which used a five-point Likert scale, mailed to patients after discharge. The sample consisted of hospitals in the northern, central, and southern regions of New Jersey. The results indicated that patient satisfaction could be a factor in the volume of teaching hospitals. However, there is an additional opportunity for improvement if teaching hospitals focus on patient satisfaction. Shahian et al. [35] examined if there is a significant difference in performance between teaching and non-teaching hospitals. This study tested the significance of performance trends of teaching intensity by analyzing categorical variables with the Mantel–Haenszel Chi-square test. The results indicated that teaching hospitals should maintain costly advances in technology to be able to specialize in patient services. In turn, teaching hospitals offer expertise in diverse and complex conditions that they know how to treat or are familiar with the procedures that should be performed. Wallace et al. [36] studied the impact of teaching hospitals on hospital management diversity. Surveys were circulated to hospital executives receiving responses from 202 of 580 surveys. The findings showed that leaders in urban teaching hospitals had a more diverse and productive team.

Based on these arguments, it is assumed that a hospital’s teaching status (teaching versus non-teaching) can control the effect of racial diversity on hospital operational efficiency. The following hypothesis is thus proposed for empirical testing. 

**Hypothesis** **5.***Hospital teaching status will control the impacts of workforce racial diversity on hospital operational efficiency*.

## 3. Methodology 

### 3.1. Data Collection and Sampling

The sample data were obtained from the American Hospital Association (AHA) Survey (n = 6232). In 2017, the AHA survey was distributed to all hospitals in the United States. Based on the AHA survey, U.S. hospitals are divided into two groups: the high-diversity group (Group 1) and the low-diversity group (Group 0). Foreign nurse employment (FORNRSA) in this survey is used as a proxy variable to measure the level of the ethnic diversity of the workforce in healthcare settings. A total of 694 U.S. hospitals were classified as Group 1, and a total of 3176 U.S. hospitals as Group 0. As a result, the effective data sample size actually used for hypothesis testing in this study was 3870 US hospitals. Additional hospital financial data (n = 6261 U.S. hospitals) were obtained from the Healthcare Cost Report Information System (HCRIS) dataset collected by the AHA to measure hospital operational effectiveness. 

This study employs the T-test and Mann–Whitney test to analyze the sample data to explore if there are significant differences between the high-diversity and low-diversity groups regarding various hospital efficiency measures. In this study, multivariate regression analysis was also considered, but the information provided by each hospital was often incomplete. That is, there were many missing values in the AHA dataset. As a result, if all control variables were added into one multivariate regression, the sample size would be drastically reduced, which would be detrimental to examining the original research question. Therefore, this study analyzed the sample data using the T-test and the Mann–Whitney test to be faithful to the original research purpose while maximizing the available sample data. In particular, we adopted Watkins’ [37] instrument variables to measure hospital operational efficiency. A more specific description of each hospital efficiency measure follows.

### 3.2. Staff Efficiency (STAFF_E)

We computed staff efficiency as one of the measures of hospital efficiency. This variable was based on full-time equivalent employees (FTEs) over the number of occupied beds (FTEs/BED) used in a study by Watkins [37]. More specifically, like Watkins [37], this variable was calculated by taking the number of fully employed workers and services provided for inpatients, then dividing this number by the annual average of occupied beds.

### 3.3. Capacity Productivity (CP)

We calculated capacity efficiency based on the study by Watkins [37]. Watkins [37] defined capacity productivity as a measure of inpatient activity produced by each bed and overall productivity across hospitals. Watkins [37] calculated this variable by multiplying the admissions by the case mix. This calculation determined the level of total inpatient activity (CMAAD). With this calculation, Watkins [37] determined bed turnover by dividing CMAAD by the number of beds in service or capacity. 

### 3.4. Manpower Productivity (MP)

We also computed manpower productivity based on the study of Watkins [37]. The study [37] utilized full-time equivalent employees (FTEs) as an indication of hospital input. The ratio was calculated by dividing the CMAAD by FTEs. 

### 3.5. Occupancy Rate (OCCP)

In Watkins [37], the variable occupancy percentage (OCCP) or occupancy rate was also proposed to measure the hospital’s efficiency. In the study [37], this variable measured hospital utilization based on its existing capacity. Hence, the occupancy rate was computed by dividing the patient days by the number of beds in service times 365.

### 3.6. Case Mix Index (CMI)

We also computed the case mix index (CMI) that Watkins [37] proposed. In Watkins [37], the CMI was used to determine the average relative diagnosis-related group (DRG) weight of the hospital’s inpatient discharges. In Watkins [37], the CMI measured the severity of hospital services based on the perception of patients treated. The case mix index was then utilized to adjust the admissions number in evaluating the CMAAD using the DRGs [37]. In fact, the government uses DRGs to regulate the amount of hospital reimbursement from Medicare patients [37].

## 4. Data Analysis 

### 4.1. Results on Hospital Efficiency by Diversity

The SPSS Explore function detected extreme values in each variable. We regarded them as outliers and removed them until SPSS reported no extreme values. Table 1 describes the data filtering process for each variable. Data filtering substantially improved each variable’s data distribution. The skewness and kurtosis results were between −1 and +1, indicating no outliers or abnormal distribution.

We performed the Kolmogorov–Smirnov test for normality, and the results, in Table 2, showed statistical significance, indicating that the data were not normally distributed. Thus, we employed the Mann–Whitney nonparametric statistical model to test the hypotheses.

This paper found that four out of the five efficiency measures, occupancy rate (OCCP), manpower productivity (MP), capacity productivity (CP), and case mix index (CMI), had statistical significance (*p* < 0.001). The efficiency measure that did not have statistical significance was staff efficiency (STAFF_E). The results indicated that the OCCP for hospitals with more workforce racial diversity (0.621 ± 0.176) was significantly more efficient than hospitals with less workforce racial diversity (0.541 ± 0.187) {MWU = 374,053, Z = −8.643, *p* < 0.001}, as shown in Table 3. The results demonstrate that the MP for hospitals with more workforce racial diversity (13.297 ± 4.884) was significantly more efficient than those with less workforce racial diversity (12.154 ± 4.997) {MWU = 337,066, Z = −4.461, *p* < 0.001}. A summary of the Mann–Whitney test results is shown in Table 3. The results also reveal that the CP for hospitals with more workforce racial diversity (81.819 ± 26.464) was significantly more efficient than for hospitals with less workforce racial diversity (74.351 ± 28.243) {MWU = 332,495, Z = −4.916, *p* < 0.001}, as presented in Table 3. In addition, the results show that the CMI for hospitals with more workforce racial diversity (1.685 ± 0.249) was significantly more efficient than those with less workforce racial diversity (1.609 ± 0.272) {MWU = 302,787.5, Z = −6.044, *p* < 0.001}, as shown in Table 3. 

In short, the results indicated that the following hospital operational efficiency measures: occupancy rate, manpower productivity, capacity productivity, and case mix index, were significantly (*p* < 0.001) correlated to the workforce racial diversity. Therefore, empirical evidence strongly supports Hypothesis 1, which states that workforce racial diversity is positively associated with hospital operational efficiency. In other words, the more racially diverse a hospital’s medical staff is, the more efficiently the hospital tends to operate.

### 4.2. Results on the Control Effect of Hospital Size

In our sample data, 47.5% of hospitals (n = 1171) were hospitals with a medium bed size, 36.5% of hospitals (n = 900) were hospitals with a small bed size, and only 16% of hospitals (n = 395) were hospitals with a large bed size, as shown in Table 4. The results are shown in Table 4 below. Hence, this study explores if there is any control effect of hospital size (i.e., bed size) on the impacts of workforce racial diversity on hospital operational efficiency.

The results for STAFF_E indicated that hospitals with more workforce racial diversity and a small bed size (7.219 ± 2.720) were not significantly more efficient than hospitals with less workforce racial diversity (7.248 ± 2.735) {MWU = 32,124.0, Z = −0.035, *p* = 0.486}. The results for STAFF_E for medium bed size (5.9640 ± 2.2630) also did not indicate that more diversity was not significantly more efficient than hospitals with less diversity (5.839 ± 2.207) {MWU = 111,118.0, Z = −0.787, *p* = 0.431}. In addition, the results indicated that the STAFF_E for large bed size did not show that more racial diversity (6.965 ± 2.468) was not significantly more efficient than hospitals with less racial diversity (6.799 ± 2.413) {MWU = 15,089.0, Z = −0.745, *p* = 0.228}, as shown in Table 5. 

However, the results indicated that the OCCP for hospitals with more workforce racial diversity for all bed sizes, which include small, medium, and large, was statistically significant. Hospitals with a small bed size (0.479 ± 0.205) were significantly more efficient than hospitals with less diversity (0.423 ± 0.174) {MWU = 34,911.5, Z = −2.655, *p* < 0.01}. Hospitals with a medium bed size (0.619 ± 0.143) were significantly more efficient than hospitals with less diversity (0.600 ± 0.149) {MWU = 110,410.0, Z = −1.811, *p* < 0.05}. Furthermore, the results indicated that hospitals with a large bed size (0.730 ± 0.124) were significantly more efficient than hospitals with less racial diversity (0.704 ± 0.111) {MWU = 15,503.0, Z = −2.189 *p* < 0.05}. A summary of the Mann–Whitney test results for the small, medium, and large bed sizes is shown in Table 5. The results demonstrated that the MP for hospitals with a medium bed size was more diverse (14.078 ± 4.870) and significantly more efficient than hospitals with less diversity (13.171 ± 4.747) {MWU = 100,964.0, Z = −2.843, *p* < 0.01}, as presented in Table 5. The results also revealed that the CP for hospitals with a medium bed size and more diversity (79.988 ± 25.886) were significantly more efficient than hospitals with less diversity (75.299 ± 26.619) {MWU = 105,531.0, Z = −2.091, *p* < 0.05}. The results show that the CMI for hospitals with a medium bed size was more diverse (1.6607 ± 0.1960) and significantly more efficient than those with less diversity (1.621 ± 0.212) {MWU = 99,675.5, Z = −3.193, *p* < 0.001}, as shown in Table 5.

In summary, the results found that four of the five efficiency measures, OCCP, MP, CP, and CMI, had statistical significance (*p* < 0.05). Notably, the results show that the medium bed size was consistently significant in OCCP, MP, CP, and CMI. Hence, Hypothesis 2, that hospital size will have a control effect on the impacts of workforce racial diversity on hospital operational efficiency is supported, except for the data of STAFF_E.

### 4.3. Results on the Control Effect of Hospital Location

In our sample data, 70.4% of hospitals (n = 1735) were in metropolitan locations, 18.9% were in microcities (n = 466), and only 10.7% (n = 265) of the hospital were in rural areas, as presented in Table 6. This study explores if there is any control effect by the location of hospitals in rural, microcity, or metropolitan locations.

The results also showed that workforce racial diversity matters more in hospitals located in metropolitan areas. The results revealed that the OCCP in metropolitan locations (0.655 ± 0.153) was significantly more diverse than hospitals located in microcities (0.493 ± 0.193) and rural areas (0.459 ± 0.203) {MWU = 208,286.0, Z = −7.117, *p* < 0.001}, as shown in Table 7. The results indicated that the MP of the metropolitan locations (14.075 ± 4.671) was significantly more diverse than hospitals located in microcities (9.145 ± 3.758) and rural areas (9.364 ± 4.428) {MWU 226,363.0, Z = −2.592 *p* < 0.05}. The results showed that the CP in metropolitan locations (86.732 ± 23.671) was significantly more diverse than hospitals located in microcities (56.870 ± 24.648) and rural areas (42.590 ± 25.304) {MWU = 222,921.0, Z = −3.134, *p* < 0.001}. The results also demonstrated that the CMI in metropolitan locations (1.734 ± 0.224) was significantly more diverse than hospitals located in microcities (1.444 ± 0.209) and rural areas (1.366 ± 0.260) {MWU = 197,271.0, Z = −4.585, *p* < 0.001}, as presented in Table 7. However, the results for the STAFF_EA located in metropolitan areas indicated (6.451 ± 2.369) that diversity was not significant. So, again, excluding STAFF_EA, all other efficiency measures in hospitals in metropolitan areas showed significantly higher diversity scores.

To sum up, the occupancy rate (OCCP), manpower (MP), and case mix index (CMI) showed more racial diversity in metropolitan areas resulting in significantly higher diversity scores than in the small cities and rural areas. The results suggest that workforce racial diversity matters more if hospitals are in a metropolitan location, while workforce diversity may not make much difference in microcities and rural hospitals. Thus, Hypothesis 3 is supported.

### 4.4. Results on the Control Effect of Hospital Ownership

In our sample data, 71.6% of hospitals (n = 1765) were non-government-owned and not-for-profit hospitals, 14.6% of hospitals (n = 359) were owned by the government, and only 13.9% of hospitals (n = 342) were investor-owned as well as for-profit hospitals, as demonstrated in Table 8. This study explored if there was any control effect of hospital ownership on the relationship between workforce racial diversity and hospital operational efficiency.

The results for the STAFF_E indicated that hospitals with more racial diversity for NG-NFP (6.904 ± 2.376) were significantly more efficient than hospitals with less racial diversity (6.631 ± 2.449) {MWU = 204,898.5, Z = −2.173, *p* < 0.05}. The results for the STAFF_E for For-Profit hospitals indicated that more diversity (4.384 ± 1.844) was significantly more efficient than For-Profit hospitals with less diversity (5.132 ± 2.262) {MWU = 8174.5, Z = −3.092, *p* < 0.01}, as shown in Table 9. The results indicated that the OCCP for hospitals with more diversity for all ownership or legal entities, which includes Govt, NG-NFP, and For-Profit. Govt hospitals (0.596 ± 0.197) were significantly more efficient than hospitals with less diversity (0.519 ± 0.211) {MWU = 7170.0, Z = −2.599, *p* < 0.01}. NG-NFP hospitals (0.648 ± 0.164) were also significantly more efficient than hospitals with less diversity (0.563 ± 0.178) {MWU = 181,874.5, Z = −8.211, *p* < 0.001}. Furthermore, the results indicated that For-Profit hospitals (0.528 ± 0.174) were significantly more efficient than hospitals with less diversity (0.451 ± 0.178) {MWU = 8302.0, Z = −3.596 *p* < 0.001}. In addition, the results demonstrated that the MP for hospitals with more diversity for all ownership or legal entities, which includes Govt, NG-NFP, and For-Profit, was statistically significantly significant. Govt hospitals (10.358 ± 4.012) were significantly more efficient than hospitals with less diversity (8.955 ± 3.700) {MWU = 4787.0, Z = −2.091, *p* < 0.05}. NG-NFP hospitals (12.973 ± 4.635) were significantly more efficient than hospitals with less diversity (12.293 ± 4.760) {MWU = 184,331.0, Z = −2.470, *p* < 0.01}. For-Profit hospitals (16.654 ± 4.673) were also significantly more efficient than hospitals with less diversity (14.341 ± 5.755) {MWU = 7292.0, Z = −3.095 *p* = 0.001}. Furthermore, the results revealed that the CP for Govt hospitals with more diversity (71.592 ± 27.405) was significantly more efficient than hospitals with less diversity (63.479 ± 29.087) {MWU = 5031.0, Z = −1.660, *p* < 0.05}. The effect of the CP of NG-NFP hospitals with more diversity (86.049 ± 25.921) was significantly more efficient than hospitals with less diversity (77.085 ± 26.812) {MWU = 162,928.0, Z = −5.269, *p* < 0.001}, as shown in Table 9. The results showed that the CMI for Govt hospitals with more diversity (1.637 ± 0.286) was significantly more efficient than hospitals with less diversity (1.521 ± 0.277) {MWU = 4128.0, Z = −2.474, *p* < 0.01}. The effect on CMI for NG-NFP hospitals with more diversity (1.7048 ± 0.2429) was significantly more efficient than hospitals with less diversity (1.619 ± 0.259) {MWU = 152,783.5, Z = −5.852, *p* < 0.001}, as shown in Table 9. 

In short, the results indicated that all five hospital efficiency measures, STAFF_E, OCCP, MP, CP, and CMI, had statistical significance (*p* < 0.05) and effect based on the ownership or legal entity (OLE). Accordingly, Hypothesis 4, that hospital ownership will control the impacts of workforce racial diversity on hospital operational efficiency, is supported.

### 4.5. Results on the Control Effect of Hospital Teaching Status

In our sample data, 91.4% of hospitals (n = 2254) were non-teaching, and only 8.6% (n = 212) were teaching hospitals, as shown in Table 10 below. This analysis explores if there is any control effect on the teaching hospital status.

The results demonstrated that the OCCP for non-teaching hospitals with more racial diversity (0.596 ± 0.172) was significantly more efficient than teaching hospitals with more racial diversity (0.768 ± 0.116) {MWU = 305,635, Z = −7.267, *p* < 0.001}, as shown in Table 11. The results indicated that the non-teaching MP for hospitals with more diversity (13.560 ± 5.055) was significantly more diverse than teaching hospitals with more diversity (11.879 ± 0.3.541) {MWU 251,879.0, Z = −5.018, *p* ≤ 0.001}. The results indicated that the non-teaching CP for hospitals with more diversity (78.435 ± 26.336) was significantly more diverse than teaching hospitals with more diversity (100.183 ± 0.18.494) {MWU 265,677.00, Z = −3.740, *p* ≤ 0.001}. The results indicated that the non-teaching CMI for hospitals with more diversity (1.644 ± 0.227) was significantly more diverse than teaching hospitals with more diversity (1.957 ± 0.212) {MWU 249,159.5, Z = −5.097, *p* ≤ 0.001}. However, the results for STAFF_E indicated that non-teaching hospitals with more diversity (6.204 ± 0.407) were not significantly more efficient, and neither were teaching hospitals (8.215 ± 2.172) {MWU 341,292.5, Z = −1.1070, *p* ≤ 0.143}, as shown in Table 11. 

In conclusion, the results revealed that diversity does matter in non-teaching hospitals. Apart from the efficiency measure of Staff_E, all other efficiency measures (OCCP, MP, CP, and CMI) in non-teaching hospitals showed statistical significance (*p* < 0.001) compared to teaching hospitals. The results implied that in non-teaching hospitals, more workforce racial diversity may lead to more hospital efficiency, while workforce diversity may not make much difference in teaching hospitals. Therefore, Hypothesis 5, that hospital teaching status will have a control effect on the impacts of workforce diversity on hospital operational efficiency, is strongly supported, except for the data of STAFF_E, by empirically showing that workforce diversity is more positively related to hospital efficiency in non-teaching hospitals than in teaching hospitals.

## 5. Discussion

### 5.1. Implications for Theory

This study empirically investigated the effect of the racial diversity of medical staff on hospital operational efficiency. The test results supported the hypotheses proposed in this study. Above all, the sample data of this study showed that workforce ethnic diversity in hospitals is significantly associated with efficiency variables, including the occupancy rate (OCCP), manpower productivity (MP), capacity productivity (CP), and case mix index (CMI). Accordingly, the empirical results strongly support Hypothesis 1 (H1), that workforce racial diversity is positively associated with hospital operational efficiency. This result suggests that hospitals tend to be more efficient with racially diverse staff. These results align with previous literature showing that diverse perspectives and ideas based on management diversity can help organizations drive more thoughtful strategies and decision-making, ultimately improving operational efficiency and profitability [38,39,40]. The study results also agree with the previous finding of Singal [14] that improving overall workplace diversity has a positive impact on financial performance in the hospitality industry. The study results are also in line with the study of Ye et al. [15] that gender diversity was positively correlated with financial efficiency.

In addition, this study investigated which factors were influential as control variables in the relationship between workforce racial diversity and hospital operational efficiency. Regarding Hypothesis 2 (H2) testing, the results indicated that the medium size hospital was consistently significant in OCCP, MP, CP, and CMI. In other words, the results demonstrated that as the racial diversity of medical staff increases, medium-sized hospitals (100 to 399 beds) are generally more advantageous in terms of operational efficiency than small-sized (<100 beds) or large-sized (≥400 beds) hospitals in the U.S. Concerning Hypothesis 3 (H3), the OCCP, MP, and CMI showed significantly higher diversity scores in metropolitan areas than in small cities and rural areas. This finding suggests that workforce racial diversity matters more if hospitals are in a metropolitan location, while workforce racial diversity may not make much difference in microcity and rural hospitals. The results of Hypothesis 4 (H4) testing indicated that the hospital ownership status also significantly controlled the relationship between workforce racial diversity and hospital efficiency. Finally, for Hypothesis 5 (H5), the results showed that workforce racial diversity had a more positive effect on hospital efficiency in non-teaching hospitals than in teaching hospitals.

Productivity is basically the concept of output divided by inputs. Thus, if equal inputs (but higher racial diversity) lead to better hospital operating outcomes, then innovative inputs (i.e., higher racial diversity) can be considered a factor positively impacting productivity. As such, the findings of this study provided indirect but empirical support for the mechanism of “why” and “how” the DEI association is linked to productivity. However, compared to other manpower-oriented service industries (education, software, banking, etc.), the hospital industry has extremely severe legal and institutional restrictions on hiring foreign medical staff. As a result, from an economic point of view, the elasticity of labor supply and demand is not sensitive to changes and is very low. These structural limitations are, thereby, the most significant barrier to effectively implementing DEIs in healthcare organizations.

The study’s findings can be supported by the contingency theory and stakeholder theory. First, the results showed that the severe shortage of medical staff due to environmental conditions could be solved by expanding the recruitment of foreign medical staff, that is, by expanding the racial diversity of medical staff. These findings are consistent with the contingency theory perspective that survival and sustainable competitive advantage could be secured when an organization adapts its organizational structure by responding appropriately to environmental changes [19,21]. Further, organizational behavior that enhances hospital operation efficiency by improving the racial diversity of hospital staff can be supported by a stakeholder approach that promotes the mutual interests and support of internal and external stakeholders [20,23,24].

This study also made some contributions regarding research methodology for measuring hospital efficiency. A conventional approach to measuring hospital efficiency is primarily linked to financial health. On the other hand, in this study, the staff efficiency, capacity productivity, manpower productivity, occupancy rate, and case mix index were adopted from the operations management perspective. Accordingly, instead of measuring financial health (e.g., cost reduction and revenue increase), this study measured how much hospitals can improve their human and material resource capabilities and, thereby, operational efficiency. As such, we expect this study also contributes to research methodology by attempting to measure the intrinsic value of foreign medical staff by evaluating the impact on operational process efficiency rather than conventional financial results.

### 5.2. Implications for Practice

First, this study aimed to explore the relationship between the racial diversity of medical staff and hospital efficiency. That is, the primary purpose of this study was to answer the following research question: *how does the racial diversity of medical staff affect hospital operational efficiency?* Test results indicated a significant relationship between workforce racial diversity and the efficiency of hospital operations. So, the implications for hospital management are clear, strongly supporting H1: hospitals tend to be more efficient when they have an ethnically diverse staff. This H1 is the core test of this study, and the remaining H2 (hospital size), H3 (hospital location), H4 (hospital ownership), and H5 (teaching status) look at which structural factors can influence the H1 relationship. Test results showed that the impact of the racial diversity of medical staff on operational efficiency was enhanced when hospitals met certain operational conditions (e.g., medium-sized, metropolitan location, and non-teaching status).

However, these findings do not imply that hospital operating conditions must be artificially altered to strengthen the H1 relationship. Instead, these findings suggest that hospitals with these specific operating conditions may need to pay more attention to racial diversity in the workplace, as they are structurally more sensitive to the relationship between racial diversity and operational efficiency. For example, hospitals in large cities are expected to have more racially diverse patients than hospitals in smaller cities. In other words, it is expected that there are more patients of various races in large cities than in small cities. Therefore, this study’s results imply that medical staff from various racial backgrounds are more likely to contribute to the efficiency of hospital operations in large cities than in small cities.

Next, employees’ innovative behavior is fundamental to an organization’s sustainable competitiveness [41]. Employees’ innovative behaviors can be further stimulated and activated in high-quality workplace relationships where diverse backgrounds and ideas are mutually respected [41,42]. An effective workforce is crucial to success and enables organizations to achieve their goals in a rapidly changing business environment [43]. Especially, due to global supply chain disruptions in the COVID-19 pandemic, organizational flexibility and resilience have become more critical factors than ever for corporate survival [44,45,46]. McKinsey’s research argued that companies that invest in diversity can gain a competitive advantage by building a more adaptable and innovative workforce [1]. The findings of this study empirically supported this argument as well.

Lastly, a previous study [11] reported that foreign-born workers comprise about one-sixth of the U.S. nursing workforce, and the need for them was soaring during the COVID-19 crisis, as reported by nursing associations and workforce agencies. In particular, our study used foreign nurse employment (FORNRSA) as a proxy variable to measure the level of ethnic diversity in hospital personnel. In consequence, our findings suggested that increasing the ratio of foreign nurses has a more positive effect on hospital operation efficiency than otherwise. Recently, the American Nursing Association (ANA) urged the U.S. Department of Health and Human Services to declare the nurse shortage a national crisis [11]. Therefore, the results of this study are expected to be one of the solutions to the ANA’s claim and further contribute to dispelling concerns and distrust regarding the recruitment of foreign nurses.

### 5.3. Limitations and Future Research Directions

This study is expected to provide meaningful insights to hospital executives, stakeholders, and healthcare policymakers by providing empirical evidence on why workforce racial diversity efforts are necessary for hospital operational efficiency. However, this study has some limitations that should be considered for future research.

First, the findings of this study could be verified using a more recent AHA dataset. That is, replication studies using the latest AHA data will be able to compare the impact of diversity on hospital efficiency before and after the COVID-19 crisis.

Second, the extended model could test additional factors such as organizational culture, leadership style, mission statement, and diversity training program as control variables and can also include other dependent variables such as the average length of stay (ALOS) and patient satisfaction to measure different angles of hospital efficiency.

Third, this study employed the T-test and Mann–Whitney nonparametric model to analyze the data to see if there were significant differences regarding hospital efficiency measures between the high-diversity and low-diversity groups. Future studies could employ multivariate regression analysis and structural equation modeling as alternative methods for analysis.

Fourth, since this study only targeted hospitals in the U.S., there may be limitations in generalizing to other countries or healthcare settings. The findings of this study should only be interpreted in the context of U.S. hospitals. We thus suggest that future studies validate the findings of this study using hospital samples from other countries and further explore how differences in healthcare systems and environments across countries affect the relationship between medical staff diversity and hospital efficiency.

Finally, this study used the foreign nurse (FORNRSA) pool as a proxy for hospital staffing racial diversity. However, MD diversity (i.e., foreign-trained medical doctors) is also expected to impact hospital operational efficiency positively. In a similar vein to this claim, previous research has pointed to a positive relationship between board racial diversity and corporate reputation and innovation [47]. Therefore, future research is suggested to explore the relationship between MD or leadership diversity and hospital operational efficiency.

## 6. Conclusions

This study aims to answer the research question of how workforce racial diversity affects hospital operational efficiency. The empirical evidence from this study strongly supports a positive relationship between the racial diversity of medical staff and the efficiency of healthcare operations in the context of U.S. hospitals. The findings of this study suggest that hospital efficiency can be operationally improved by implementing workforce racial diversity. For this reason, hospital stakeholders and healthcare policymakers are expected to benefit from this study’s findings. In fact, investors and stakeholders have been increasingly interested in ESG management [16,24]. Socially conscious investors look for companies that embody an ethical and socially conscious work environment, from subcontractors to top management [17,18], and these claims can be supported by the stakeholder theory [22,23].

As such, seeking racial diversity in the workforce is now seen as a fundamental barometer of organizational behavior that predicts successful ESG business practices. In particular, the COVID-19 pandemic has forced healthcare to innovate to adapt to the changing public health landscape [48]. The severe nurse shortage caused by the COVID-19 pandemic also requires changes in federal healthcare policy to be more flexible and inclusive in hiring foreign medical staff. In this context, the findings of this study provide additional evidence for increasing funding to strengthen DEI policies in the healthcare industry.

Therefore, in conclusion, we propose that the racial diversity of the workforce should no longer be an optional consideration but should be considered as one of the essential determinants of competitive advantage in organizations, particularly in the healthcare sector [49,50].

## Figures and Tables

**Figure 1 behavsci-13-00564-f001:**
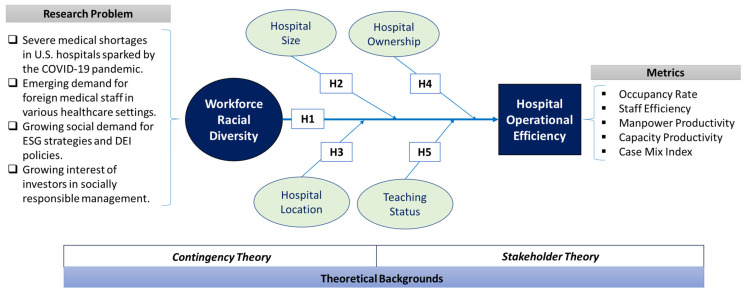
Research framework.

**Table 1 behavsci-13-00564-t001:** The data filtering of efficiency measures.

Variable	Filtering Criteria	Before Filtering			Outliers	After Filtering		
		N1	SK	KT		N2	SK	KT
STAFF_E	STAFF_E < 13.2	2466	2.112	9.123	168	2298	0.504	−0.293
OCCP	OCCP < 1.11	2466	0.086	0.554	8	2458	−0.143	−0.512
MP	MP ≤ 26.3	2135	0.987	2.727	33	2102	0.348	−0.418
CP	CP ≤ 158	2135	0.716	2.764	35	2100	−0.092	−0.320
CMI	0.87 ≤ CMI ≤ 2.33	2135	0.860	1.892	82	2053	0.212	−0.200

**Note:** STAFF_E = staff efficiency, OCCP = occupancy percentage, MP = manpower productivity, CP = capacity productivity, CMI = case mix index, SK = skewness, KT = kurtosis, N1 = initial sample size, and N2 = sample size after filtering the data.

**Table 2 behavsci-13-00564-t002:** The descriptive statistics and normality test results.

Variable	Mean	Median	SD	Min	Max	Skewness	Kurtosis	DF	K-S Statistic
STAFF_E	6.494	6.185	2.519	1.22	13.16	0.504	−0.293	2298	0.053 ***
OCCP	0.558	0.572	0.187	0.059	1.075	−0.143	−0.512	2458	0.038 ***
MP	12.414	12.053	4.994	0.588	26.294	0.348	−0.418	2104	0.043 ***
CP	76.065	77.803	28.015	4.488	153.107	−0.092	−0.320	2100	0.032 ***
CMI	1.627	1.615	0.269	0.917	2.329	0.212	−0.200	2053	0.022 *

**Note:** * *p* < 0.05, *** *p* < 0.001, STAFF_E = staff efficiency, OCCP = occupancy percentage, MP = manpower productivity, CP = capacity productivity, CMI = case mix index, K-S Statistics = Kolmogorov–Smirnov test statistic for normality.

**Table 3 behavsci-13-00564-t003:** The Mann–Whitney test results on hospital operational efficiency by workforce racial diversity.

Efficiency	Diversity	N	Mean	SD	MR	MWU	Z	*One-Tailed p*
STAFF_E	Less	1813	6.497	2.532	1148.54	437,910.5	−0.134	0.447
	More	485	6.481	2.472	1153.09			
OCCP	Less	1947	0.541	0.187	1166.12	374,053.0	−8.643	<0.001
	More	511	0.621	0.176	1471.00			
MP	Less	1625	12.154	4.997	1020.43	337,066.0	−4.461	<0.001
	More	479	13.297	4.884	1161.31			
CP	Less	1618	74.351	28.243	1015.00	332,495.0	−4.916	<0.001
	More	482	81.819	26.464	1169.68			
CMI	Less	1585	1.609	0.272	984.03	302,787.5	−6.044	<0.001
	More	468	1.685	0.249	1172.52			

**Note:** SD = standard deviation, MWU = Mann–Whitney test statistic, STAFF_E = staff efficiency, OCCP = occupancy percentage, MP = manpower productivity, CP = capacity productivity, and CMI = case mix index.

**Table 4 behavsci-13-00564-t004:** U.S. hospital size.

	Frequency	Percent	Valid Percent	Cumulative %
Small (6–99 beds)	900	36.5	36.5	36.5
Medium (100–399 beds)	1171	47.5	47.5	84.00
Large (≥400 beds)	395	16.0	16.0	100.00
Total	2466	100.0	100.0	

**Table 5 behavsci-13-00564-t005:** The control effect of hospital size on workforce racial diversity and hospital efficiency.

Efficiency	Bed Size	Diversity	N	Mean	SD	MR	MWU	Z	*p*
STAFF_E	Small	Less	685	7.248	2.735	390.10	32,124.0	−0.035	0.486
		More	94	7.219	2.720	389.24			
	Medium	Less	890	5.839	2.207	570.35	111,118.0	−0.787	0.431
		More	258	5.964	2.263	588.81			
	Large	Less	238	6.799	2.413	182.90	15,089.0	−0.745	0.228
		More	133	6.965	2.468	191.55			
OCCP	Small	Less	791	0.423	0.174	440.14	34,911.5	−2.655	0.004
		More	105	0.479	0.205	511.51			
	Medium	Less	906	0.600	0.149	575.37	110,410.0	−1.811	0.035
		More	263	0.619	0.143	618.19			
	Large	Less	250	0.704	0.111	187.51	15,503.0	−2.189	0.015
		More	143	0.730	0.124	213.59			
MP	Small	Less	501	9.565	4.651	285.93	17,499.0	−1.312	0.095
		More	77	10.119	4.223	312.74			
	Medium	Less	875	13.171	4.747	553.39	100,964.0	−2.843	0.002
		More	261	14.078	4.870	619.16			
	Large	Less	249	13.796	4.544	197.74	16,996.0	−0.522	0.301
		More	141	13.587	4.575	191.54			
CP	Small	Less	491	64.294	30.054	281.54	17,449.0	−0.910	0.182
		More	76	66.852	27.619	299.91			
	Medium	Less	877	75.299	26.619	559.33	105,531.0	−2.091	0.018
		More	263	79.988	25.886	607.74			
	Large	Less	250	90.777	20.882	193.60	17,026.0	−0.784	0.217
		More	143	93.142	21.859	202.94			
CMI	Small	Less	481	1.454	0.281	279.38	17,613.5	−0.328	0.372
		More	75	1.426	0.226	272.85			
	Medium	Less	871	1.621	0.212	550.44	99,675.5	−3.193	<0.001
		More	263	1.661	0.196	624.01			
	Large	Less	233	1.887	0.210	182.15	15,109.0	−0.038	0.485
		More	130	1.885	0.192	181.72			

Note: *p* = one-sided *p*-value, SD = standard deviation, MR = mean rank, MWU = Mann–Whitney U test statistic, bed size small (beds ≤ 99), medium (100 to 399 beds), large (beds ≥ 400), STAFF_E = staff efficiency, OCCP = occupancy percentage, MP = manpower productivity, CP = capacity productivity, and CMI = case mix index.

**Table 6 behavsci-13-00564-t006:** U.S. hospital location.

	Frequency	Percent	Valid Percent	Cumulative %
Rural	265	10.7	10.7	10.7
Microcity	466	18.9	18.9	29.6
Metropolitan	1735	70.4	70.4	100.00
Total	2466	100.0	100.0	

**Table 7 behavsci-13-00564-t007:** The control effect of hospital location on workforce racial diversity and hospital efficiency.

Efficiency	Location	Diversity	N	Mean	SD	MR	MWU	Z	*p*
STAFF_E	Rural	Less	205	6.606	3.0687	112.59	1966.0	−0.654	0.257
		More	21	7.066	2.861	122.38			
	Microcity	Less	347	6.728	2.805	209.94	11,473.5	−0.546	0.293
		More	69	6.482	2.911	201.28			
	Metropolitan	Less	1261	6.416	2.347	826.22	246,174.5	−0.346	0.365
		More	395	6.450	2.369	835.77			
OCCP	Rural	Less	238	0.447	0.199	130.62	2647.0	−0.591	0.278
		More	24	0.459	0.203	140.21			
	Microcity	Less	389	0.455	0.166	229.15	13,283.0	−1.399	0.081
		More	76	0.493	0.193	252.72			
	Metropolitan	Less	1320	0.584	0.176	818.29	208,286.0	−7.117	<0.001
		More	411	0.655	0.153	1019.22			
MP	Rural	Less	92	7.311	3.096	50.14	335.0	−1.407	0.080
		More	10	9.364	4.428	64.00			
	Microcity	Less	304	8.839	3.610	184.08	9601.0	−0.547	0.292
		More	66	9.145	3.758	192.03			
	Metropolitan	Less	1229	13.338	4.819	799.18	226,363	−2.592	0.005
		More	403	14.075	4.671	869.31			
CP	Rural	Less	92	40.990	24.776	50.79	395.0	−0.226	0.411
		More	9	42.590	25.304	53.11			
	Microcity	Less	304	55.666	22.941	184.72	9796.0	−0.300	0.382
		More	66	56.870	24.648	189.08			
	Metropolitan	Less	1222	81.510	25.649	793.92	222,921.0	−3.134	0.001
		More	407	86.732	23.671	878.28			
CMI	Rural	Less	91	1.2652	0.1592	49.69	336.0	−1.353	0.176
		More	10	1.366	0.260	62.90			
	Microcity	Less	304	1.447	0.200	186.31	9786.0	−0.312	0.378
		More	66	1.444	0.209	181.77			
	Metropolitan	Less	1190	1.676	0.257	761.27	197,271.0	−4.585	<0.001
		More	392	1.734	0.224	883.26			

Note: *p* = one-sided *p*-value, SD = standard deviation, MR = mean rank, MWU = Mann–Whitney U test statistic, NT = non-teaching hospitals, STAFF_E = staff efficiency, OCCP = occupancy percentage, MP = manpower productivity, CP = capacity productivity, and CMI = case mix index.

**Table 8 behavsci-13-00564-t008:** U.S. hospital ownership.

	Frequency	Percent	Valid Percent	Cumulative %
Government (Govt)	359	14.6	14.6	14.6
Non-government, not-for-profit (NG-NFP)	1765	71.6	71.6	86.1
Investor Owned, for-profit (For-Profit)	342	13.9	13.9	100.00
Total	2466	100.0	100.0	

**Table 9 behavsci-13-00564-t009:** The control effect of hospital ownership on workforce racial diversity and hospital efficiency.

Efficiency	OLE	Diversity	N	Mean	SD	MR	MWU	Z	*p*
STAFF_E	Govt	Less	266	7.086	2.744	161.85	7541.5	−0.267	0.395
		More	58	7.154	2.233	165.47			
	NG-NFP	Less	1305	6.631	2.449	810.01	204,898.5	−2.173	0.015
		More	340	6.904	2.376	872.86			
	For-Profit	Less	242	5.132	2.262	174.72	8174.5	−3.092	0.001
		More	87	4.384	1.844	137.96			
OCCP	Govt	Less	298	0.519	0.211	173.56	7170.0	−2.599	0.005
		More	61	0.596	0.197	211.46			
	NG-NFP	Less	1395	0.563	0.178	828.38	181,874.5	−8.211	<0.001
		More	362	0.648	0.164	1074.08			
	For-Profit	Less	254	0.451	0.178	160.19	8302.0	−3.596	<0.001
		More	88	0.528	0.174	204.16			
MP	Govt	Less	209	8.955	3.700	127.90	4787.0	−2.091	0.019
		More	56	10.358	4.012	152.02			
	NG-NFP	Less	1185	12.293	4.760	748.55	184,331.0	−2.470	0.007
		More	341	12.973	4.635	815.44			
	For-Profit	Less	231	14.341	5.755	147.57	7292.0	−3.095	0.001
		More	82	16.654	4.673	183.57			
CP	Govt	Less	210	63.479	29.087	129.46	5031.0	−1.660	0.049
		More	56	71.592	27.405	148.66			
	NG-NFP	Less	1183	77.085	26.812	729.72	162,928.0	−5.269	<0.001
		More	339	86.049	25.921	872.39			
	For-Profit	Less	225	70.122	31.732	153.58	9131.0	−0.919	0.179
		More	87	71.921	23.626	164.05			
CMI	Govt	Less	200	1.521	0.277	121.14	4128.0	−2.474	0.007
		More	53	1.637	0.286	149.11			
	NG-NFP	Less	1173	1.619	0.259	717.25	152,783.5	−5.852	<0.001
		More	330	1.705	0.243	875.52			
	For-Profit	Less	212	1.638	0.317	146.60	8500.5	−0.762	0.223
		More	85	1.639	0.241	154.99			

Note: *p =* one-sided *p*-value, SD = standard deviation, MR = mean rank, MWU = Mann–Whitney U test statistic, OLE = ownership control/legal entity, Govt = government hospitals, NG-NFP = non-government and not-for-profit hospitals, STAFF_E = staff efficiency, OCCP = occupancy percentage, MP = manpower productivity, CP = capacity productivity, and CMI = case mix index.

**Table 10 behavsci-13-00564-t010:** U.S. hospital teaching status.

	Frequency	Percent	Valid Percent	Cumulative %
Teaching	212	8.6	8.6	8.6
Non-teaching	2254	91.4	91.4	100.0
Total	2466	100.0	100.0	

**Table 11 behavsci-13-00564-t011:** The control effect of hospital teaching status on workforce racial diversity and hospital efficiency.

Efficiency	Teaching	Diversity	N	Mean	SD	MR	MWU	Z	*p*
STAFF_E	Teaching	Less	123	8.1201	2.275	94.08	3946.0	−0.482	0.315
		More	67	8.215	2.172	98.10			
	NT	Less	1690	6.379	2.510	1061.55	341,292.5	−1.1070	0.143
		More	418	6.204	2.407	1025.99			
OCCP	Teaching	Less	136	0.747	0.109	102.24	4588	−1.354	0.088
		More	76	0.768	0.116	114.13			
	NT	Less	1811	0.526	0.183	1074.77	305,635	−7.267	<0.001
		More	435	0.596	0.172	1326.39			
MP	Teaching	Less	136	12.262	3.974	108.68	4736.0	−0.857	0.196
		More	75	11.879	3.541	101.15			
	NT	Less	1489	12.145	5.082	914.16	251,879.0	−5.018	<0.001
		More	404	13.560	5.055	1068.04			
CP	Teaching	Less	133	97.869	18.299	101.17	4544.0	−1.064	0.144
		More	75	100.183	18.494	110.41			
	NT	Less	1485	72.244	28.024	921.91	265,677.0	−3.740	<0.001
		More	407	78.435	26.336	1036.23			
CMI	Teaching	Less	115	1.945	0.2487	88.75	3536.5	−0.088	0.465
		More	62	1.957	0.212	89.46			
	NT	Less	1470	1.583	0.256	905.00	249,159.5	−5.097	<0.001
		More	406	1.644	0.227	1059.81			

**Note:** *p* = one-sided *p*-value, SD = standard deviation, MR = mean rank, MWU = Mann–Whitney U test statistic, NT = non-teaching hospitals, STAFF_E = staff efficiency, OCCP = occupancy percentage, MP = manpower productivity, CP = capacity productivity, and CMI = case mix index.

## Data Availability

Data presented in this study are available from the American Hospital Association Data and Insight.

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
