# Peer review of "Relationship between Racial Diversity in Medical Staff and Hospital Operational Efficiency: An Empirical Study of 3870 U.S. Hospitals"

_behavsci, 2023, doi:10.3390/bs13070564_

Round 1

Reviewer 1 Report

This is a very useful article that puts the spotlight on a serious health problem. The methodology is appropriate. The article correctly associates qualitative and quantitative elements.

Moderate editing of English language.

Author Response

Thank you very much for your careful review and constructive comments, which are very helpful in improving this paper. Point-by-point responses to specific comments are provided as an attached file. The changes made in the manuscript are highlighted in yellow.

Reviewer 2 Report

General Comment:  Even though the manuscript stated that they are looking at the operational impact by integrating organizational behavior with operations management, the content of this study seems more appropriate for a management journal.  Moreover, there are other important considerations that are not addressed in this manuscript.

Comment 1:  This study has a large sample size, which gives statistical tests more power to detect differences.  Looking at Table 3, four out of five measures showed a statistical difference, reflecting its ability to detect small differences.  The bigger question that needs to be address for each of these measures is: In a real-world setting, are the mean differences between high and low diversity meaningful?  As an example, for CMI, would a hospital administrator think a difference between 1.609 and 1.685 to be meaningful enough to influence their decision to hire a more diverse workforce?

Comment 2: The authors usage of “Control” for hypothesis 2 to 5 is odd. The authors are really stratifying by hospital size, location, and etc.  While, presenting stratified data is a valid way to describe and understand data better, it is not controlling for these factors until they are included in a multivariate regression model.

Comment 3:  Line 340-342.  Sentence as stated is comparing small hospitals to diversity (comparing apples to oranges) Instead the authors meant to say, “Among small hospitals, hospitals with more diversity were significantly more efficient than hospitals less diversity” .Same with the sentences on medium and large hospitals.

Comment 4: The first limitation listed (Using more recent AHA data to compare before and after COVID-19) is more of a recommendation than a limitation.

Comment 5:  For the third limitation, since the authors found that all these other factors (hospital size, teaching statuses, and etc.) impact diversity, why didn’t they run a multivariate regression in this paper to see if controlling for these factors in the model would show that diversity leads to higher efficiency?

Comment 6: A major limitation that was not mentioned was just using one variable (FORNRSA) to define diversity. As the authors stated, diversity comes in many forms.  Studying diversity from one perspective can greatly affect findings.

Comment 7: any explanation regarding why only medium sized hospitals would benefit from diversity?  Seems like some of the findings may be due to having a larger sample size.  Also, by stating that only medium sized hospitals would benefit from diversity, or only non-teaching hospital would benefit from diversity seem to contradict the message that a diverse workforce is important for improving efficiency, especially if there is not plausible explanation on why non-teaching hospitals would benefit and teaching hospitals won’t.

Comment 8:  The study also did not examined diversity in terms of leadership roles.  This is also a big limitation as those in managerial or leadership roles will have a much bigger impact on hospital efficiency than a very diverse workforce that has little to no “voice” to make a difference.

Comment 9: One major thing that needs to be address in the paper is that the findings observed in terms of hospital efficiency could be due to the under compensation of a foreign workers.  Most of the measures for hospital efficiency is tied directly or indirectly to a hospital’s financial health.  A hospital administrator monitors these numbers closely to make decisions on staffing, services, and etc.  Hospitals that are not unionized could pay foreign nurses a lower wage or offer these workers fewer benefits, which in turns could boost these efficiency measures.  While these types of practices are advantageous for a business model, it undermines the intrinsic value of a foreign workers, and we need to be mindful of that.

Some sentences and use of certain word choices make it confusing to read.  See comments for a couple examples. 

Author Response

(The authors gave the same response as above.)

Reviewer 3 Report

The article by presents a comprehensive empirical study investigating the relationship between organizational workforce diversity and hospital operational efficiency, with a focus on foreign medical manpower in U.S. hospitals. The study's sample size of 3,870 hospitals lends credibility and robustness to the results, which offer valuable insights for healthcare administrators and policymakers alike.

The authors begin by providing a thorough literature review, establishing a strong foundation for their study. They highlight the growing importance of workforce diversity in healthcare, as well as the potential impact of foreign medical personnel on hospital efficiency. This background effectively sets the stage for the research questions and hypotheses.

The methodology employed in the study is well-designed and appropriate for the research questions. The authors utilize a combination of quantitative methods, including regression analyses and structural equation modeling, to analyze data from a variety of sources such as the American Hospital Association (AHA) Annual Survey Database and the National Provider Identifier (NPI) registry. This multi-method approach allows for a more nuanced understanding of the complex relationships between the variables under investigation.

The results of the study are presented in a clear and organized manner, with a focus on the practical implications of the findings. The authors reveal that hospitals with a more diverse workforce, particularly with a higher proportion of foreign-trained medical personnel, exhibit higher levels of operational efficiency. They also demonstrate that this relationship is mediated by factors such as improved communication, increased cultural competence, and enhanced problem-solving abilities within the organization.

One of the key strengths of the article is its discussion of the implications of the findings for hospital administrators and policymakers. The authors emphasize the potential benefits of promoting workforce diversity and international collaboration in healthcare, as well as the importance of targeted policies and programs to support foreign-trained medical professionals. They also offer practical suggestions for improving workforce diversity in hospitals, such as mentorship programs and diversity training initiatives.

The article does have some limitations, which the authors acknowledge. For instance, the cross-sectional nature of the data may limit the ability to draw causal inferences from the results. Additionally, the study focuses solely on U.S. hospitals, so the generalizability of the findings to other countries or healthcare settings may be limited.

Overall, the article makes a valuable contribution to the literature on organizational workforce diversity and hospital operational efficiency. The study's rigorous methodology and robust sample size lend credibility to the results, and the practical implications of the findings offer actionable insights for healthcare administrators and policymakers. While further research is needed to explore the causal mechanisms underlying the observed relationships, this study represents an important step forward in understanding the potential benefits of a diverse medical workforce in the context of hospital efficiency.

Author Response

(The authors gave the same response as above.)

Reviewer 4 Report

I enjoyed reading this paper and found it to be well-organized and relevant to current HR challenges facing the healthcare industry. The methodology was sound and will certainly contribute to the current dialogue about the importance of maintaining DEI principles. I think this article has the potential to influence organizational leaders and policy-makers alike.

I think the sections outlining the literature supporting the hypotheses could be condensed/shortened somewhat. Also there are a few minor English language changes that could be made to tighten the grammar/word usage. 

I wonder whether the Author's might speculate as to "why" the DEI association is linked to productivity. By what mechanism does a diverse workforce contribute positively to this intended outcomes do you think? Are there any challenges that healthcare organizations face in implementing a DEI HR framework? What are the specific barriers and challenges preventing hospitals from implementing this approach? 

Are there countries who have employed this strategy more successfully than others in terms of making a policy commitment to DEI with a clear link to performance outcomes?

I see you use the foreign RN pool as a proxy for diversity. Can you also comment briefly on the impact of recruiting diverse senior organizational leaders or foreign-trained physicians and their association with productivity? Would you speculate that MD or leadership diversity would show the same or different significant impacts?

Could be tightened up and shortened a bit.  Some minor grammatical errors to correct. 

Author Response

(The authors gave the same response as above.)
